

# Electric resistivity and seismic refraction tomography, a challenging joint underwater survey at Äspö Hard Rock Laboratory

Mathias Ronczka[1], Kristofer Hellman[1], Thomas Günther[2], Roger Wisen[1], and Torleif Dahlin[1]

[1]Lund University, Lund, Sweden
[2]Leibniz Institute for Applied Geophysics, Hannover, Germany

*Correspondence to:* Mathias Ronczka (mathias.ronczka@tg.lth.se)

**Abstract.** Tunnelling below water passages is a challenging task in terms of planning, pre-investigation and construction. Fracture zones in the underlying bedrock lead to low rock quality and thus reduced stability. For natural reasons they tend to be more frequent at water passages. Ground investigations that provide information of the subsurface are necessary prior to the construction phase, but can be logistically difficult. Geophysics can help close the gaps between local point information and produce subsurface images. An approach that combines seismic refraction tomography and electrical resistivity tomography has been tested at the Äspö Hard Rock Laboratory (HRL). The aim was to detect fracture zones in a well-known but logistically and, from a measuring perspective, challenging area.

The presented surveys cover a water passage along a part of a tunnel that connects surface facilities with an underground test laboratory. The tunnel is approximately 100 m below and 20 m east of the survey line and gives evidence for one major and several minor fracture zones. The geological and general test site conditions, e.g. with strong powerline noise from the nearby nuclear power plant, are challenging for geophysical measurements. Co-located positions for seismic and ERT sensors and source positions are used on the 450 m long underwater section of the 700 m long profile. Because of a large transition zone that appeared in the ERT result and the missing coverage of the seismic data, fracture zones at the southern and northern part of the underwater passage cannot be detected by separated inversion. A simple synthetic study shows significant three dimensional artefacts corrupting the ERT model that have to be taken into account while interpreting the results. A structural coupling cooperative inversion approach is able to image the northern fracture zone successfully. In addition, previously unknown sedimentary deposits with a significant large thickness are detected in the otherwise unusually well documented geological environment. The results significantly improve imaging of some geologic features, which would have been not detected or misinterpreted otherwise, and combines the images by means of cluster analysis to a conceptual subsurface model.

## 1 Introduction

Underground structures have become an increasingly important part in modern infrastructure, and the possibilities to improve construction approaches have attracted much attention. With constantly reduced space for new structures on the surface, the underground space is attractive to use in the transportation sector to challenge the growth of traffic in and around cities, or for underground storage facilities. Geological uncertainties increase the risk of delays and thus costs for underground constructions.





A detailed subsurface model is essential for reducing the risks and for a successful project. A critical point in order to ensure a smooth construction phase is to locate present weak zones, and especially those that can generate large inflow of water causing problems and slowing down the construction progress. Except for southwestern Scania and the islands Gotland and Öland, crystalline bedrock is the dominating material for underground infrastructure construction in Sweden. For these geologic conditions, weakness zones that are important for the underground design are normally indicated by dry, water-bearing or sediment-filled fractures.

Two methods for site investigation in crystalline bedrock are drilling and surface-based or borehole geophysics. Drilling is often the first choice since it gives high resolution and accuracy at any given depth. Nevertheless, drilling is expensive and delivers only point information. Therefore, surface-based geophysical methods have gained more attention, since they give continuous models that reveal the extreme points and give an opportunity for extrapolation into 2D or 3D space. Recently, the Swedish transportation authority has provided funding for research in an increasing number of projects with the aim to develop site investigations based on additional geophysical measurements for mapping of the structure and quality of the rock mass.

Dahlin et al. (1999) reports on a case where electrical resistivity tomography (ERT) has been used successfully for mapping weak and permeable rock in an on shore railway tunnel project in Sweden. Ha et al. (2010) used different geoelectrical applications to detect weak zones of approx. 40 m × 40 m during an underground construction. In the Norwegian R&D project „Tunnels for the citizens" funded by the road administration, several publications (Karlsrud et al., 2003; Palmstrøm et al., 2003; Rønning et al., 2013; Wisén et al., 2012; Lindstrøm and Kveen, 2004) report that elaborate site investigations are important in a controlled tunnelling process, but also that further studies are needed. Rønning et al. (2013) assessed ERT, refraction seismics, very low frequency (VLF) electromagnetics and the AMAGER-method (Aeromagnetics and Geomorphological Relations) and concluded that they are all able to locate fracture zones. They state that ERT is able to give more hints to the fracture width, dip and depth extent compared to the other methods used. They also suggest a quantitative rock quality measure on the base of resistivity values. Refraction seismics has been since long an established method to give information on fracture width and seismic p-wave velocity, the latter having an obvious coupling to the hardness of the rock and hence to rock quality (Bergman et al., 2006). Diaz et al. (2014) successfully conducted seismic refraction and ERT surveys and associated resistivity and velocity changes with main and secondary structures of a major fault zone. Final velocity and resistivity models were also consistent with deformed sedimentary units. Another multidisciplinary geophysical approach for mapping a fault zone is given in Malehmir et al. (2016). Heincke et al. (2010) used seismic and electric tomography to assess the rock quality on a hardrock slope in Norway. Repeatedly, several methods are combined to overcome the limits of the natural resolution and corresponding ambiguity in inversion and interpretation. One example of for synthetic and field data is given in Garofalo et al. (2015), where seismic data and ERT were used to reduce model ambiguities and improve the estimation of geophysical parameter.

This paper describes a field case where Seismic and ERT surveys were conducted at Äspö Hard Rock laboratory (HRL). The main objective was the localisation and characterisation of fracture zones under challenging test site conditions, because a water passage was crossed. Dahlin and Wisén (2016) and Günther and Südekum (2007) showed that underwater field surveys are possible and quite promising. In order to increase the reliability of the results, a combined inversion and interpretation of both methods was investigated. This was done by joint inversion followed by a cluster analysis as an additional integrated




interpretation approach. After describing the site conditions and the numerical background, we show a synthetic study on 3D effects before we actually analyse and interpret the field data.

## 2 Site description

The Swedish Nuclear Fuel and Waste Management Company (SKB) started to design a deep final disposal for nuclear fuel.
Äspö HRL is SKB's underground facility for research and tests of a concept for final disposal of nuclear waste material in hard rock (Rhén et al., 1997). The laboratory has provided a test environment in full scale for different technological solutions. It has now mainly filled its purpose so that the laboratory has become available also for other branches of research. The facility provides a research opportunity in a well-documented and relatively undisturbed geological environment that is representative for many Swedish metropolitan areas.

The Äspö Hard Rock Laboratory is located on the east coast of the Baltic Sea, about 400 km south of Stockholm (see Figure 1). From 1990 to 1995 the excavation of a 3600 m long tunnel that connects the nuclear power plant with the disposal in approximately 450 m depth was conducted. During the construction phase, a detailed site characterization was done that included geological, hydrogeological and geochemical investigations.

[Figure 1 about here.]

The Äspö bedrock is part of the Trans-Scandinavian Igneous Belt (TIB) that extends from southern Sweden towards North and Northwest. Generally, granitoids and volcanic rocks can be found in the TIB. Four rock types are dominating: the Äspö diorites, Ävrö granite, greenstone and fine-grained granite. Wikberg et al. (1991) found out that continuous magma mixing processes supported the development of dikes and mafic inclusions which form an inhomogeneous rock mass. The crystalline bedrock exhibits porosities of 0.4-0.45% for the Äspö diorite and 0.23-0.27% for the fine-grained granite (Stanfors et al.,
1999). During the pre-investigation of Äspö HRL, fracture zones were divided into major (width > 5 m) and minor (width < 5 m) ones. The majority of the fractures are oriented northwest-southeast (Berglund et al., 2003). All fracture zones that are important for this field survey are depicted as black lines in Figure 1.

Filling material of the fractures was extracted from drill cores and analysed. Missing unconsolidated material that might have been additionally filling the fractures was probably washed away and thus not taken into account in these analysis. Calcite
crystallised in the fractures was possibly formed by hydrothermal processes and can be used as an indicator for water paths in the rock (Wikberg et al., 1991). This indicated that fractures in N-S and E-W directions most likely conduct or formerly conducted water. Four zones are of interest for this field case: NE-1 crossing the northern part of the profile and the fracture zones NE-3, NE-4 and EW-7, which cross the southern part of the conducted seismic and ERT profile. Wikberg et al. (1991) also stated that quaternary sediments on top of the bedrock were supposed to be scarce at the Äspö test site. Due to the deep
target of the Äspö HRL within the bedrock, no detailed investigation of the Quaternary sediments was done. Vidstrand (2003) stated that the unconsolidated overburden should rarely exceed 5 m thickness and consists mainly of clay, sand and gravel.



## 2.1 Electrical resistivity tomography

ERT measurements were carried out along a profile in N-S direction, simultaneously with the seismic survey, on April 20-24, 2015. The profile lies between Hålö and Äspö (see Figure 1) to the west of the tunnel line, about 10 m away from a small island. Electrodes were placed onshore and underwater, with a 5 m electrode spacing along a 780 m long profile. Data were

recorded using the multi-channel instrument ABEM Terrameter LS. A multiple gradient array (Dahlin and Zhou, 2006) was employed to ensure fast measuring progress as it can fully exploit the recording channels.

The site conditions were challenging because of the fact that well-defined (and co-located with seismics) sensor positions had to be ensured for an underwater survey. Additionally, a near by power plant caused a high noise level in the ERT data. Large variations of the contact impedance between the water part and rock outcrops gave a technically difficult measuring situation.

Contact resistances, including cable resistance, started from 100 Ω for electrodes in brackish water and exceed 100 kΩ on rock outcrops. The full wave form of the transmitted and received signals was recorded in order to recover possibly valuable IP signals from the data. However, the signal-to-noise ratio was sufficiently good for recovering DC resistivity but not IP data. About 6700 data points were gathered during the ERT survey. While processing the raw data, electrodes with an apparently wrong GNNS position were identified and all combinations containing these electrodes were deleted. To account for the

variable data quality of the individual data, usually a data error is estimated by a fixed percentage and a voltage error. They can be retrieved by analysing reciprocal measurements Udphuay et al. (2011), which were however not available here. Therefore we used the default values of 3% noise and a voltage error of 0.1 mV.

## 2.2 Seismic refraction tomography

The green dashed line in Figure 1 marks the profile for the seismic refraction. Hydrophone streamers were laid out with 91

hydrophones in total and a 5 m spacing along a 450 m profile line. For data acquisition the instruments ABEM Terraloc and Geometrics Stratavizor were used, both with 48 channels and with a 5 channel overlap of the two streamers. Hydrophone positions were determined by a differential GNSS, while the topography of the sea bed was mapped with a multibeam echo sounder. For the excitation of seismic p-waves, small explosives were placed approximately 0.5 m above the sea bed. Shots were performed every 20 m. Due to time constraints, not all planed shots were fired and hence there are two small gaps in the

data coverage in the northern part of the dataset. Raw data processing revealed that the seismic signal quality was significantly reduced in the southern part of the profile, which made it difficult to pick first arrivals. However, no additional filters were used during the raw data processing. About 650 first arrival times were semi-automatically picked and manually checked using the software package Rayfract (www.rayfract.com).

## 3 Numerical modelling and inversion

We used the open-source ERT software packages BERT (Boundless Electrical Resistivity Tomography) for ERT inversion (Günther et al., 2006b) using irregular triangle meshes to take into account both surface and submarine topography accurately





Rücker et al. (2006). Furthermore, we used the underlying framework pyGIMLi (Python Geophysical Inversion and Modelling Library, www.pygimli.org) for the refraction tomography and the implementation of the coupled inversion.

## 3.1 Inversion

Inversion of both ERT and SRT was done by a smoothness-constrained minimisation with the cost function

$$\Phi = \Phi_d + \lambda\Phi_m \tag{1a}$$

$$= \sum_{i=1}^{N} \left( \frac{d_i - f_i(\boldsymbol{m})}{\epsilon_i} \right)^p + \lambda ||\boldsymbol{C}\boldsymbol{m}||_p^p) \tag{1b}$$

containing an error-weighted data misfit $\Phi_d$ and a model roughness $\Phi_m$ weighted by the regularisation parameter $\lambda$. The model parameters are logarithmic resistivities, held in the model vector $\boldsymbol{m}$. The difference between the individual data points $d_i$ and the corresponding forward responses $f_i(\boldsymbol{m})$, both as logarithmic apparent resistivities, is weighted by their individual errors $\epsilon_i$. Different norms can be used for data misfit such as the L$_2$-norm ($p = 2$) or the L$_1$-norm ($p = 1$) for an inversion that is more robust with respect to outliers (Claerbout and Muir, 1973) in case of difficult data quality. The roughness (second term in eq. 1b) consists of the derivative matrix $\boldsymbol{C}$ applied to the model $\boldsymbol{m}$ (Günther et al., 2006b). Additional model constrains can be incorporated in the object function by extending $\Phi_m$ to the weighted model functional Rücker (2011)

$$\Phi_m = ||\boldsymbol{W}_c\boldsymbol{C}\boldsymbol{m}||_p^p \tag{2}$$

The weighting matrix $\boldsymbol{W}_c$ is diagonal and contains the elements $w_i^c$ representing penalty factors for the different model cell boundaries (Günther et al., 2006a). Very small values can lead to to sharper boundaries. The limited amount and quality of recorded data leads to a non-unique inversion result. Due to the model smoothing, needed for mixed determined problems, it is possible that sharp boundaries appear as transition zones that lead to misinterpretations. A structurally coupled joint inversion finds common structures and allows the models to emphasize these and reduce smoothing effects (Gallardo and Meju, 2004). Here, the roughness vector $\boldsymbol{r} = \boldsymbol{C}\boldsymbol{W}_m\boldsymbol{m}$ is used to calculate the mutual penalty factors $w_i^c$ using a coupling equation (Günther et al., 2010). Differently from the latter approach, we multiply the $w_i^c$ of the different methods and calculate one weighting matrix for both methods.

A certain number of separated iterations is done before the coupling starts so that each method can first independently develop structures before their similarity is promoted. A schematic sketch of the structurally coupled joint inversion is shown in Figure 2.

[Figure 2 about here.]

Forward modelling and inversion are done on unstructured finite element (FE) meshes that allow to incorporate both surface as well as underwater topography accurately. All shown inversion results are faded out using the coverage to point out the con-tribution of model parts to the data. The calculation of the coverage is based on the sensitivity, which is the partial derivative





$S_{i,j}(m^n) = \frac{\partial f_i(m^n)}{\partial m_j}$. Whereas $m = \log \rho$ are model parameter and $f = \log \rho_a$ the forward response (both logarithmic transformed). The summation of all sensitivities for each model parameter gives the coverage for the model cell assigned with this parameter. The finite element mesh used for the joint inversion of seismic and ERT data is shown in Figure 3.

[Figure 3 about here.]

The shown mesh consists of three regions that present the background (red), the water (blue) and the parameter domain (green), on which the data inversion is conducted. The original mesh extension is 1250 m in x- and approximately 420 m in z-direction and is clipped for display reasons. In-situ water conductivity measurements showed resistivity values of about 1.4 Ωm and negligible variation with position or depth. As velocity of pure water is as well constant (about 1400 m/2), the water region can be assumed homogeneous and is incorporated as a single region with a fixed resistivity or velocity so that

the correct values are used for the forward calculation but are not subject to inversion. The parameter domain is extended to approximately 790 m in x- and 190 m in z-direction. Additionally, an outer background region is needed for accurate forward calculation using approximate boundary conditions Rücker et al. (2006). Although the seismic line is shorter than the ERT, the shown mesh was used for both data sets in the joint inversion. The parameter domain consists of about 3500 cells, which is the number of model parameters. More details on region-based inversion can be found in Rücker (2011).

**3.2   Synthetic study on 3D effects**

We follow a strict 2D scheme, i.e. assume constant values perpendicular to the profile. Three-dimensional (3D) effects occur, if significant resistivity changes perpendicular to a 2D profile are present. According to the test site map in Figure 1, severe 3D effects can be expected near the small island in the middle of the profile and in the northern part, where the water continues just a few meters next to the profile. The latter is not expected to have a significant effect on the first-arrival times, since

these are related to the smallest distance to the layers. It will, however, have an effect on the measured apparent resistivity by all materials present within the measured volume. In order to appraise expectable shapes and magnitudes of 3D effects, we generated a simplified model based on the Äspö geometry. The underlying model used for generating synthetic data is shown in Figure 4. The water body is simulated by a cube with an extension of 450 m in x-direction starting at x=100 m, being 10 m in depth and infinite in y-direction. A large cube simulating the bedrock (brown) surrounds the water cube, with

an infinite extension in x-, y- and z-direction. The water (blue) is assigned with a resistivity of 3 Ωm, while the bedrock is assigned with 3000 Ωm. Two anomalies are inserted representing the island in the middle and the small bay at the northern end of the ERT profile. The island (red) is a 10 m thick cube, with an extension of 90 m in x- and 70 m in y-direction, placed between x=370-460 m with a distance of 10 m to the ERT profile. The small bay at the northern part (green) is incorporated by a rectangular cube with an edge length of 100 m (x-, y-direction) and 3 m depth. It starts directly after the water cube at

x=550 m with a distance of 5 m to the profile. It is aligned along the x-direction at y=0 m, consists of 153 electrodes and starts at x=15 m according to the field survey. The simulated survey is identical to the field measurements except that the electrodes are assumed at the surface and topography is neglected. The ERT profile is marked with red spheres in Figure 4.

[Figure 4 about here.]



For reference, we additionally calculated data from a 2D model, where the island is assigned with the water resistivity of $3\,\Omega\text{m}$ and the bay with $3000\,\Omega\text{m}$ (bedrock), i.e. with no 3D effects. Both data sets were corrupted with Gaussian nose of the above described an error level consisting of of 3% plus a voltage error of $100\,\mu\text{V}$. A smoothness-constrained inversion was performed to estimate resistivity models from the two synthetic data sets. Figure 5(a) and Figure 5(b) are showing the inversion

results from the data set without and with 3D effects. The ratio between those two is shown in Figure 5(c).

[Figure 5 about here.]

Figure 5(a) shows the expected smooth resistivity distribution, with a horizontal interface between the simulated bedrock and water. By including the island and the small bay into the underlying model, serious 3D effects occur. These lead to higher resistivities in the middle of the profile, where the island was included and additional low resistive compensation artefacts next

to it. The small water filled bay at the end of the profile leads to a characteristic low resistive feature at intermediate depths. Both anomalies, including the possible compensation artefacts are more visible in the ratio plot given in Figure 5(c).

This simple synthetic study validates that the ERT data gathered at the Äspö test site are contaminated/distorted by 3D effects that have to be taken into account when interpreting the results.

## 4    Results

A smoothness-constrained inversion was done with the abort criterion $\chi^2 = \Phi_d/N = 1$, i.e. the data are fitted within their errors. Visual inspection of the data misfit ensured that there was no more unresolved structure. The $L_1$-norm data (robust) inversion was used to account for remained outliers in the ERT data set that lead to poor data fits. Nevertheless, the apparent resistivities cover several orders of magnitude ($3$–$47000\,\Omega\text{m}$) and extraordinarily high resistivity variations occur, which is challenging for ERT inversion. The ERT inversion result is shown in Figure 6a) using the coverage (sum of absolute Jacobian

values over all data for each model) for alpha-shading. In the middle of the profile, the penetration depth is limited due to the well-conducting water body and the anomalies below.

[Figure 6 about here.]

Outcrops of the bedrock lead to high resistivities of about $35000\,\Omega\text{m}$ at the northern and southern end of the profile. A low resistive zone appears at $x = 200$–$600\,\text{m}$, directly below the sea. The depth varies between approximately $80\,\text{m}$ at $x = 270\,\text{m}$ and

$30\,\text{m}$ at $x = 450$–$600\,\text{m}$. As such a deep weathering zone seems implausible and the resistivity is too low for usual weathering, we interpret this structure as a deep valley filled with sediments. This has not been documented by previous investigations conducted in the construction phase of the test nuclear waste disposal. The low resistive zone is extended diagonally downwards towards the north for $x > 600\,\text{m}$ at a depth range of $50$–$100\,\text{m}$. Although the coverage is getting low for this part, it is still possible that this feature indicates fractured water bearing bedrock.

Resistivities of about $500\,\Omega\text{m}$ at $x = 100$–$200\,\text{m}$ and a depth of $100\,\text{m}$ indicate a larger transition zone that continues below the sediment body. That could possibly lead to an incorrect depth of the sediment filled valley and thus bedrock interface. It also prevents any further interpretation regarding possible fracture zones.





The inversion result of the refraction seismic shown in Figure 6b) images the interface to the bedrock more accurately. However, the poor signal quality in the southern part results in a lower coverage and thus larger uncertainty. For displaying the inversion result, a standardised coverage was calculated, which is either 0 or 1, depending on whether any ray travels through a model cell or not.

According to Figure 6, the crystalline bedrock appears as a high velocity zone of about 5600 m/s. Towards the northern part, the velocity of the bedrock decreases down to 5000 m/s. At the southern part, between $x = 200\text{-}300\,\mathrm{m}$, the result shows a low velocity zone down to 60 m depth, which is extended towards the north for shallow parts of the model, above 20 m depth. This finding coincides with the low resistive part in ERT result. The sediments exhibit a minimum velocity of about 1000 m/s, which is below the velocity of water (1400 m/s). A reason could be gas contained in the sediments, which reduce the acoustic velocity for frequencies below 1 kHz, (Wilkens and Richardson, 1998). This is supported by the presence of gas bubbles raising up to the water surface during the blasting. It is assumed that the gas-bearing sediments lead to the poor data quality in the southern part by damping the seismic signals. No further low velocity zones at larger depth appear.

To summarise, a (possibly gas-bearing) sediment body could be identified, which appears as a zone of low resistivities and velocities. Furthermore, the interface towards the bedrock could be found by the joint interpretation of the separated inversion results. However, the bedrock appears with a low resistivity due to the large transition zone. Fracture zones are not visible in the separated inversion results (Figure 6), because of a low coverage in the refraction model and large transition zone in the resistivity model.

In order to improve the results and enable further interpretation, a structurally coupled joint inversion of the ERT and seismic data was performed. To ensure that common structures are present in the models, the first four iterations were done separately. A robust data fit, i.e. $L_1$-norm, was used for ERT-data inversion, while the first arrivals were fitted using the $L_2$-norm (least squares). Both data sets were fitted within their errors, i.e. with $\chi^2 = 1.1$ for ERT and $\chi^2 = 1.3$ for refraction data. In this case, the rms (root mean square) error for the first arrival-fit was about 2.4 ms. The result is shown in Figure 7.

[Figure 7 about here.]

Both models show significant changes compared to the separated inversions and allow further interpretations. Generally, most changes occur in the resistivity model, while the velocity model shows only small improvements. The low resistive zone, which corresponds to the sedimentary filled valley, appears thinner followed by a much smaller transition zone. This reduces the ambiguity in estimating the bedrock interface. The bedrock is also assigned with a higher resistivity which is more realistic as it agrees with the resistivity of near-surface rock outcrops at the northern and southern end of the profile.

Additional structural constraints that moved from the velocity to the resistivity model pointed out the diagonal low resistive zone in the northern part more detailed. This anomaly matches very well with the water bearing fracture zone NE-1 in the northern part of the profile. The southern fracture zones NE-3, NE-4 and EW-1 cannot be identified directly. Possible explanations could be that these are (i) too small to be detected from the surface or (ii) filled with a material so that no parameter contrast appears.





According to the synthetic study the low resistive feature directly at the surface at x=610 m and to a small part the diagonal low resistive zone at x=600 m are most likely caused by 3D effects and should not be interpreted any further. Following Günther et al. (2006a), a post-processing of the two inversion models was done by using a cluster analysis to obtain a simplified result (Figure 8). For clustering the resistivities and velocities a modified Mean Shift algorithm approach was used, which is described

in Comaniciu and Meer (2002). The input for this algorithm is a feature space, which consists in this case of resistivities and velocities. In order to analyse the feature space, a window or bandwidth is needed. The bandwidth can be determined by using a bandwidth estimator that uses a selected quantile as input. The quantile is defined between zero and one. In general, a low quantile will produce a larger number of clusters than a high quantile. In opposite to cluster number driven algorithms such as the K-means algorithm (see Joydeep and Alexander (2009)),the input is data and a window to the data. Therefore, the selection

of clusters is driven only by data and not by an arbitrary number of clusters.

As data input for the clustering, we only used model parameters included by the coverage of the seismic result (displayed cells in Fig. 7b), because the seismically covered volume is also covered by ERT.

The data driven cluster algorithm divided the model parameter were into three clusters that represent sedimentary deposits, the bedrock and the transition zone between those two. It can most likely be assumed that the interface between the sediments

and the bedrock is within the third cluster.

[Figure 8 about here.]

As a final interpretation of the presented ERT and seismic results a conceptual model was developed (Figure 9). The primary origin of the deep sedimentary deposits, can be explained by glacial erosion. The small valley was formed between the fracture zones NE-3 and NE-4. It might have been easier to erode the bedrock along zones with an already low rock quality. Two

possible explanations can be given for the remaining transition zone at the bottom of the sedimentary valley. The first one is that the bedrock-sediment interface is (i) fractured/weathered to a certain extent or (ii) that coarse sediments could have been deposited before fine grained marine material was sedimented above. The latter possibility is visualised by the dark-yellow and orange parts at the bottom of the valley in Figure 9. As the medium velocities north of the sedimentary valley appear slightly thicker, the most probable explanation could be weathered bedrock. During earlier investigation it was found out that the NE-1

fracture zone in the northern part of the model is water bearing at its boundaries and dry in its core due to clay deposits. Thus, it appears as a zone of lower resistivities and velocities. One possible explanation for water bearing and non-water bearing fractures could be that some of them are filled with sediments and some are not.

[Figure 9 about here.]

## 5 Conclusions and outlook

An combination refraction seismic and ERT data has been tested on an underwater profile crossing a water passage along a part of the access tunnel, which connects surface facilities with an underground test laboratory at the Äspö Hard Rock Laboratory.



The aim was to detect fracture zones in a well-known but logistically challenging area. Co-located sensor positions for ERT and seismic were used on a 450 m long underwater section of the 700 m long ERT profile.

Inversion results showed a previously unknown sediment filled valley that appeared as a zone with low resistivities and velocities. The poor coverage of the seismic model in the northern and southern part of the profile in conjunction with the

5 large transition zone of the ERT result prevent further detailed interpretations. However, the water bearing fracture zone NE-1 could be identified by the results of the structurally coupled joint inversion. The sharp bedrock interface in the seismic result constrained the ERT model such that a smaller transition zone appeared, which made the fracture zone visible. The southern fracture zones NE-3, NE-4 and EW-1 could not be detected due to of the missing parameter contrast and/or model resolution.

A synthetic study inspired by geologic conditions of the Äspö test site showed that significant three dimensional effects

are expected that contaminates the ERT data and thus influences the obtained inversion result. This was taken into account to prevent misinterpretation of the final inversion results. The evaluation shows that the used joint inversion approach, combining ERT and seismic has given very promising results due to three reasons: (i) the decreased extent of the transition zone, (ii) the more reliable interpretation of two independent parameters, and (iii) the combination of those two by a clustering approach. For the presented example, the continuous information provided by geophysics can reveal previously unknown geological features,

even in an unusually well documented geological environment and under the challenging underwater conditions.

*Acknowledgements.* Thanks to all, who participated in the field survey and made the measurements possible. Special thanks go to Marcus Wennermark, who planned the survey and was responsible for seismic instrumentation and data collection in the field campaign. We are grateful to SKB (Swedish Nuclear Fuel and Waste Management Co.) for logistic support during the field campaign. Funding which made this work possible was provided by Nova FoU, BeFo (Swedish Rock Engineering Research Foundation, ref. 314 and 331), SBUF (The De-

20 velopment Fund of the Swedish Construction Industry, ref.12718 and 12719) and Formas (The Swedish Research Council for Environment, Agricultural Sciences and Spatial Planning, ref. 2012-1931) as part of the Geoinfra-TRUST framework (http://www.trust-geoinfra.se/).





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


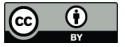

## List of Figures



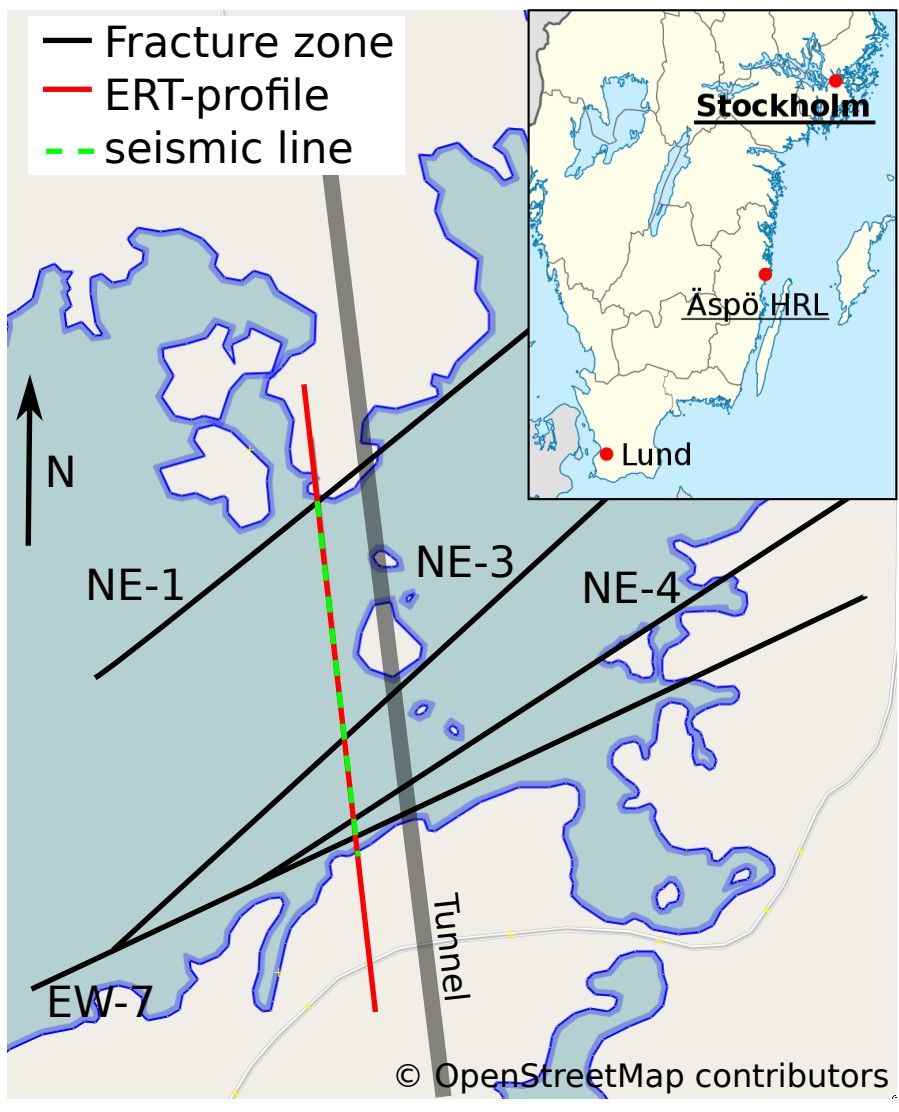

**Figure 1.** Location, major fracture zones (black lines) after Stanfors et al. (1999) and the scheduled ERT profile (solid red line) and seismic (dashed green line) at Äspö Hard Rock Laboratory.

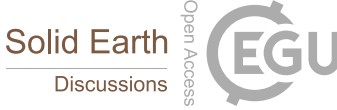



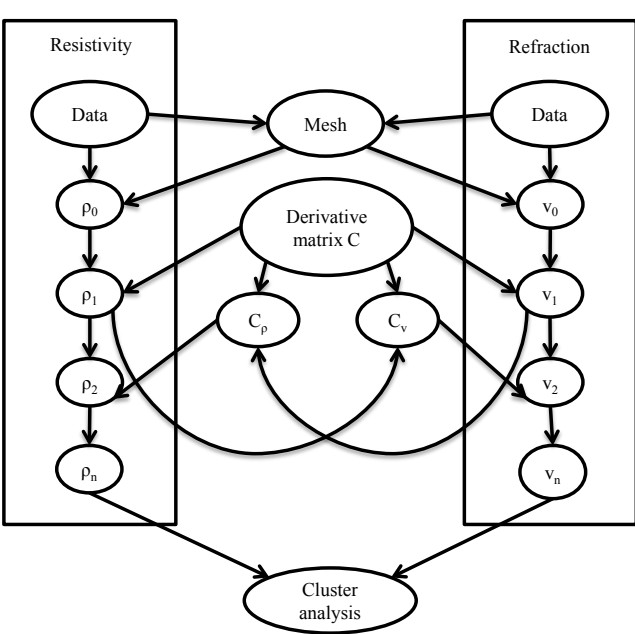

**Figure 2.** Scheme of the coupled inversion approach, where the roughness $C$ of one inversion is influenced by the other (Günther et al., 2006a).





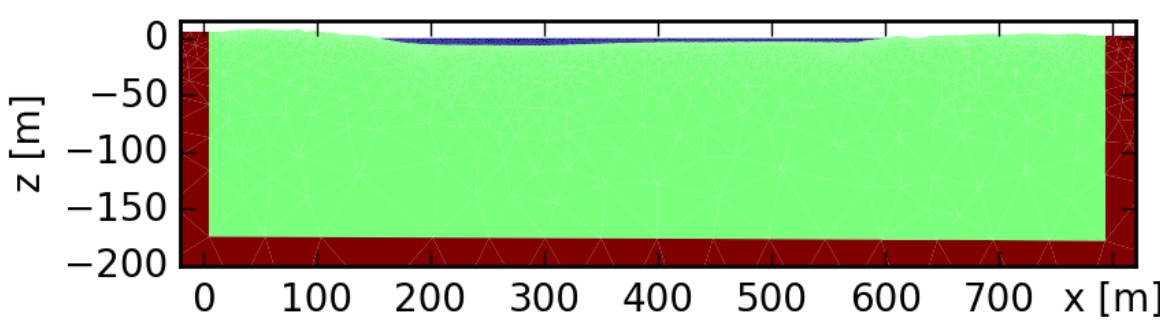

**Figure 3.** Cropped mesh used for structurally coupled inversion of ERT and seismic data. Three regions are used for (i) background in red (much bigger) to prevent influences of the boundaries, (ii) the parameter domain (green) on which the inversion is done and (ii) the water region (blue) which was fixed.





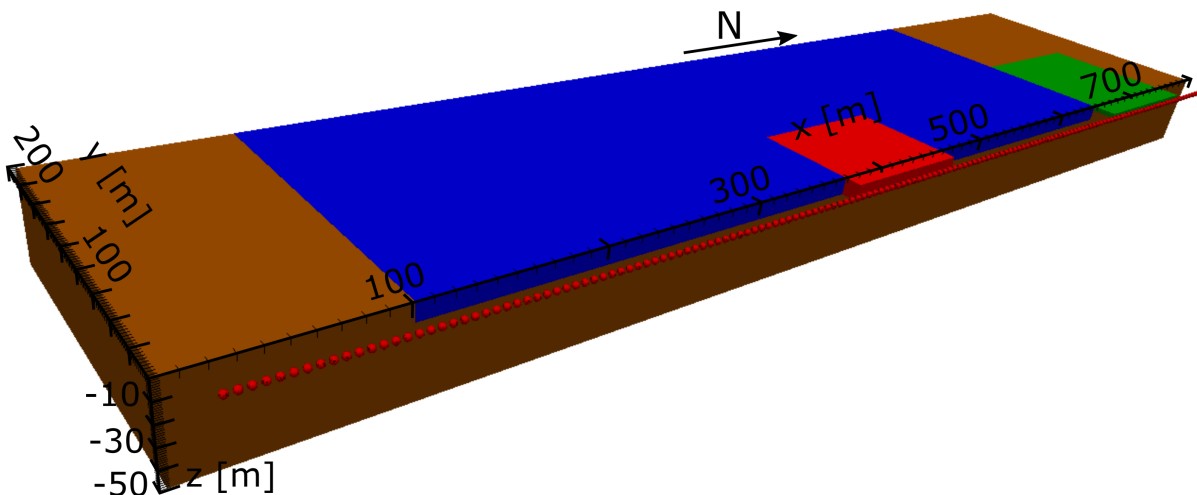

**Figure 4.** Sketch of the synthetic model used to generate synthetic data. It reflects a simplified version of the Äspö test site conditions. The red spheres mark electrode positions, blue coloured areas simulate a low resistive body, like sea water and the brown parts mark high resistive bodies, like bedrock.





**Figure 5.** Inversion results of the synthetic case with (a) a pure 2D model, (b) the incorporated island and small bay causing 3D effects and (c) the ratio between (a) and (b).







**Figure 6.** Separated inversion results of the ERT data set (a) and the refraction seismic data (b). The shading is based on the coverage.



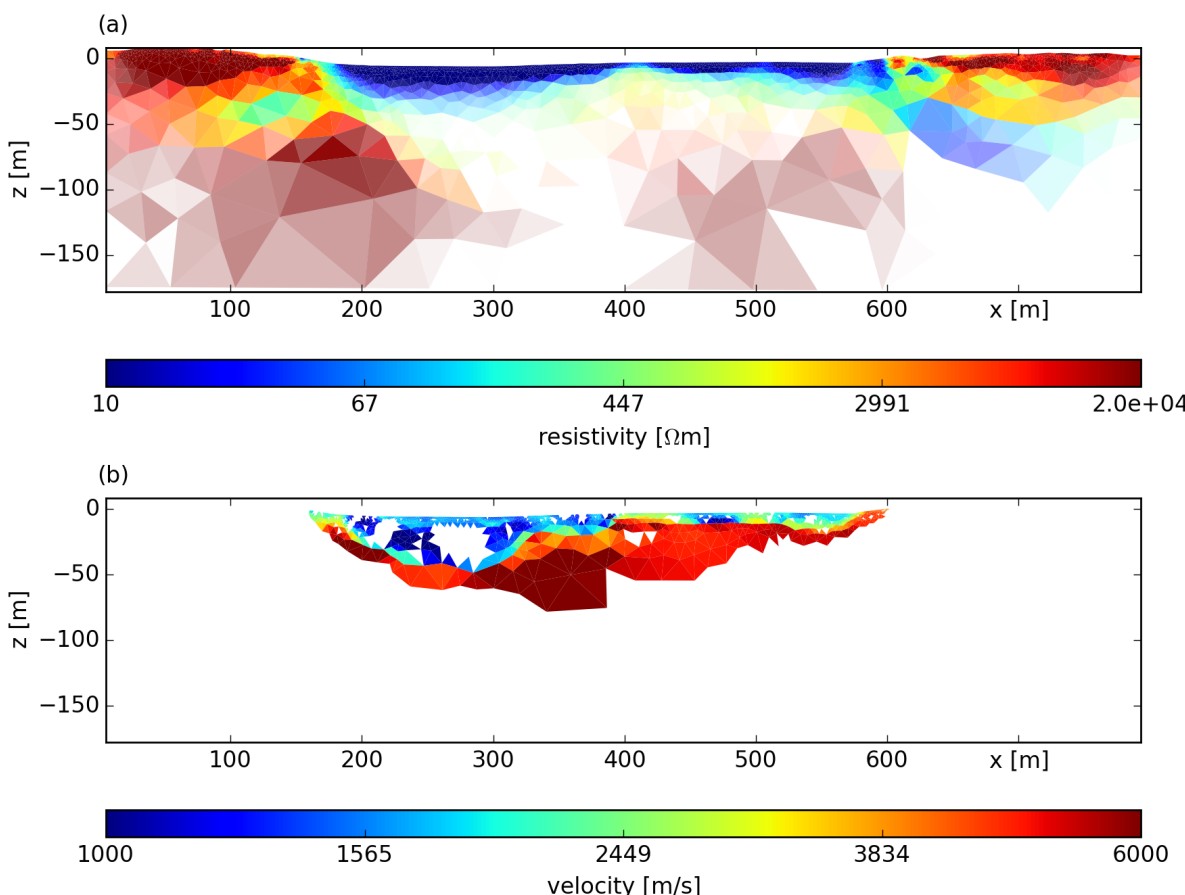

**Figure 7.** Joint inversion result with resistivity (top) and velocity (bottom) distribution. The shading is based on the coverage of each model cell.



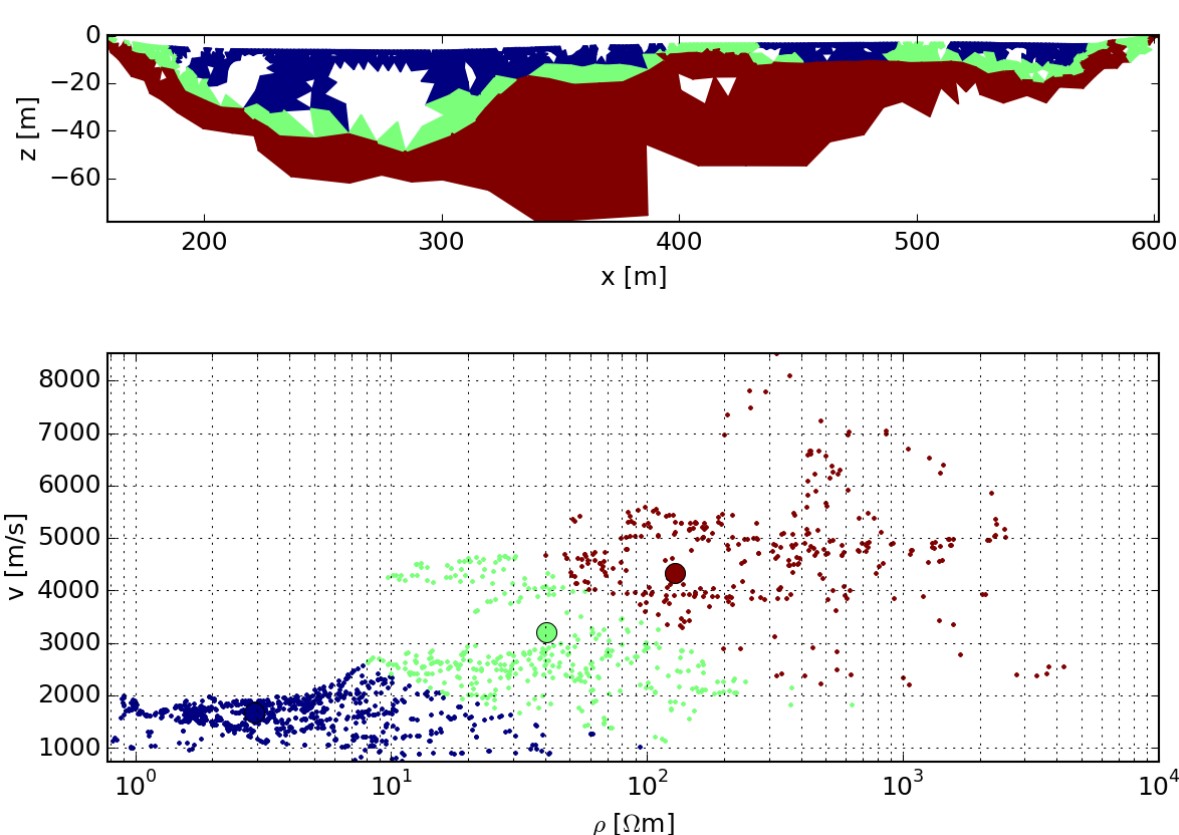

**Figure 8.** Cluster analysis of the joint inversion result using tree clusters. The upper picture shows the spatial distribution of the clusters and the lower one the parameter distribution within each cluster.





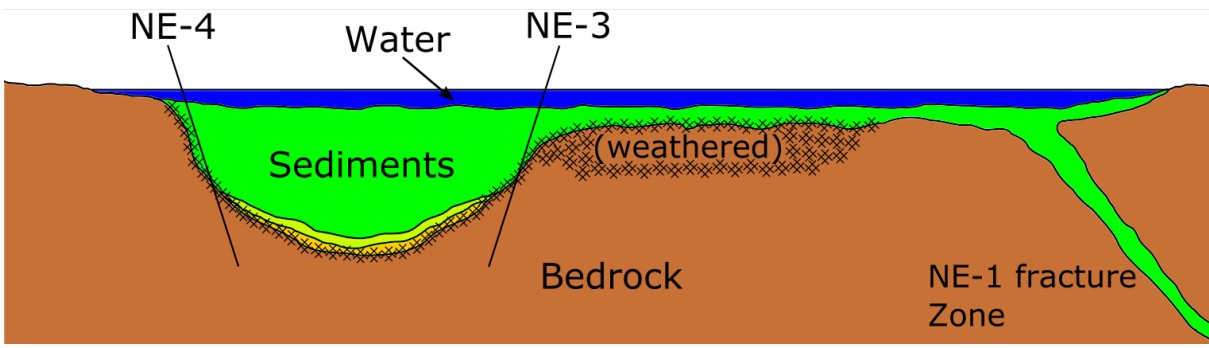

**Figure 9.** Conceptual model based on geophysical results and known geologic interpretations of the test site Äspö. The hash-signature at the bedrock interface indicates a higher uncertainty.