# Peer review of "Electric resistivity and seismic refraction tomography, a challenging joint underwater survey at Äspö Hard Rock Laboratory"

_Solid Earth, 2016_

## Referee Comment (RC1) · Anonymous Referee #1 · 1 Feb 2017

Dear Editor, dear authors, Ronczka et al. present a potentially interesting case study of combining electrical resistivity and seismic refraction tomography. Unfortunately, it is currently not clear, what practitioners and scientists can learn from their survey. The authors present data that was acquired under what they describe as difficult conditions, but it is unclear, what this really meant for the survey and results and what others should learn from it. Should seismics and ERT be standard methods to be used for site investigation under these circumstances or not?

In the introduction, the authors claim that they are testing geophysical methods for improving planning for infrastructure projects. Later, when discussing their results and conclusions, there is no mention of this main objective. Also, only one of the four

fracture zones that is geologically known was found using the survey and there is only speculation, why the others are not found. Also, it is unclear, which of the shear zones would be most critical for infrastructure projects.

The introduction speaks about some of the general difficulties of site investigations and how geophysics can help. It also names quite a few studies that have tried to use different geophysical methods in environments of crystalline rock. However, the strengths and weaknesses of the different methods in conditions of crystalline rock in combination with high-conductivity water are not clearly discussed. It is also not clearly discussed why the current study is needed and what it should add, compared to the existing ones. Generally, the introduction should summarize the current state of research and discuss gaps that are addressed in the manuscript. Here, it is not clear, which gaps are being addressed and how. Furthermore, the introduction very much targets Sweden, while the study should be of more general interest to be published in an international journal.

The site description is a very short description of all the work that has been performed at the Äspö HRL. Unfortunately, information that would be important for the current study is missing:

- What have seismic surveys in the rock lab shown?

- Was no surface seismic data acquired in the region? If other surface seismic data exists: What did the surveys find? Why are they not discussed? Just a quick search finds e.g.:

J. S. Kim, Wooil M. Moon, Ganpat Lodha, Mulu Serzu, and Nash Soonawala (1994). "Imaging of reflection seismic energy for mapping shallow fracture zones in crystalline rocks." GEOPHYSICS, 59(5), 753-765. doi: 10.1190/1.1443633

Cosma, C., Olsson, O., Keskinen, J., Heikkinen, P., 2001. Seismic characterization of fracturing at the Äspö Hard Rock Laboratory, Sweden, from the kilometer scale

to the meter scale. International Journal of Rock Mechanics and Mining Sciences, Applicationãă of Geophysics to Rock Engineering 38, 859–865. doi:10.1016/S1365-1609(01)00051-X

- Which seismic velocities are found in the rock lab for intact rock and how does it change in fracture zones?

- What is the electrical resistivity of intact rock and of fracture zones?

- How likely are sedimentary deposits in the region? Have other studies or evidence on land shown extensive sedimentary deposits?

For the measurement techniques, some problems are described, but not really discussed how to be solved. For example: contact resistances vary by a factor of more than 1000. What does this mean for the measurements? Should future survey be tried in similar areas, or rather not?

For the positions: electrode positions were apparently measured using differential GNNS. How did this work for the underwater survey? What was the water depth? Does an accurate bathymetry model exist? Was this bathymetry model used, or was the multi-beam echo sounder used? Was the multi-beam echo sounder used to measure 3D ocean bottom topography?

With a difference in resistivity of more than a factor of 10000 between the water and the rock, it seems very important to know the position of the interface accurately. Were electrode positions deleted because they were in the wrong place, or simply, because the GNNS positioning did not work accurately?

For seismic refraction tomography: Crystalline rock is a perfect target for reflection seismic surveys, as shown in many studies. Why is only refraction seismics being used here? Low velocity zones – especially if they are just thin fracture zones – are very difficult to image in refraction tomography from the surface. Reflection surveys in the area (see references above) have shown the great potential of seismic surveys under

these circumstances. If the potential of geophysical methods should be addressed here, why not process the data for both reflections and refraction and compare the results?

The section on inversion is somewhat confusing and not fully consistent: For example, Equations 1a, 1b and 2 do not agree. Using eq. 2, the weighting matrix is missing in 1b. Also, according to the text, the model vector contains logarithmic resistivities. What about the slowness or velocity? Concerning the use of different norms: Most inversion schemes (and I think the one of Günther 2006b is not different) use a reweighting scheme to implement a L1 norm. This is not a true L1 inversion and should not be confused with that. The discussion of the weighting matrix and the introduction of joint inversion on P5L15-22 is confusing and not well motivated. Why should a joint inversion be attempted in the first place? What could be the advantages here? What are the difficulties with single-method inversions that could be overcome by joint inversion? The reference of Günther et al. 2010 is rather difficult to access. Why not give the actual equation here?

Synthetic study: It is interesting, but not surprising to learn from the synthetic study that 3D effects under these circumstances can be severe. The authors conclusion is that "ERT data gathered at the Äspö test site are contaminated" and that cautions should be used for the interpretation. However, if the conclusion is that the data is contaminated, the conclusion should be that one needs to take the 3D effects into account in the 2D inversion or perform a 3D survey. Simply saying that one needs to be careful in the interpretation is something one could say even without the synthetic study. I suspect that not only the island created artefacts, but also bathymetry changes in the area of the profile effect the results. A more interesting synthetic study would be to compare the current density in the rock and in the water. With a resistivity factor of more than 10000, I would suspect that only very little current flows through the rock, which directly limits the sensitivity and makes small uncertainties in the water geometry a major problem. Would it not be possible to use a 3D forward model that incorporates the 3D bathymetry

and still invert for a 2D model along the profile? For the water, one could simply use the known water resistivity and for the rock far away from the profile, one could use the known rock resistivity. It is rather unsatisfactory to analyse a problem in a manuscript and then simply say "be careful in the interpretation", while it is unclear what "careful" really means.

The resistivity of the water seems to be rather important for the outcome of the inversion, especially if is kept fix. Would it not be possible to invers for one single water resistivity value? Is the measured resistivity that at the ambient temperature, or normalized to 25 degrees? Most conductivity meters actually give the resistivity at 25 degrees and would need to be back-corrected to the much lower water temperature. It would be interesting to see how the inversion changes when the water resistivity is kept at different (higher or lower) values and how the results change when it is left to vary freely. It might be co-incidence and actually makes geological sense, but the thick sediment deposit lies at the position with the greatest water depth and thus could be influenced by the water depth.

Results: The results are mostly a description of the ERT and refraction seismic results, without mentioning the previous studies in the area. How do the values compare to those expected from measurements at the HRL? How do they compare to the previous surface seismic lines? What else is known about the fracture zones?

In the introduction, boreholes and geophysics is given as the two most common ways to explore the subsurface. Here geophysical results are presented, but without incorporating ground truth from e.g. boreholes. One of the main findings is a thick sediment deposit. This would indeed be a nice result. But currently, the sediments are more a hypothesis than a result. Are there any other indications of this sediment deposit? Are any drill holes available in the area? From other studies in the area: Are similar deposits known to exist? Are they likely, given the geological history?

Overall, I judge that there is potential for a nice case study, but the current manuscript

can only be the starting point to shape the current manuscript into a journal publication. In my opinion, the study needs to be properly embedded in the current literature, the results – especially the ERT results – need to be improved to not include 3D artefacts and the results need to be discussed in light of local geological knowledge. The 3D synthetic study shows how severe 3D artefacts are. This also means that they need to be addressed in the inversion and a warning of "being careful" is clearly not enough. For the plausibility of the sediment hypothesis, any external verification would be important. If a borehole could demonstrate the existence and thickness of these sediments, this would indeed be a very nice result!

Technical edits:

- P3L5: The abbreviation has not been previously introduced

- P3L11: should be "disposal site"

- P3L17: Remove "out"

- P3L21: On the map, all shear zones are northeast – southwest, not northwest – southeast as in the text.

- P5L16: Remove "to".

Figure 1: There is no scale in the Figure.

Figure 2: Grid cells are partly visible, but not clearly. Either show them or remove them.

Figure 4: The red and green areas are not discussed in the caption. Why not give the resistivity values of each area in the caption?

---

## Referee Comment (RC2) · Anonymous Referee #2 · 6 Feb 2017

Dear Editor/Authors

This is a very interesting paper which uses an exemplary methodological approach for interpreting real data coming from different geophysical methods. The use of state-of-art interpretation software with custom forward modelling and combined inversions is certainly the way forward to increase the quality of geophysical interpretations in complex environments.

As I view it, the paper is very good but on the other hand it has some weak points that need to be addressed prior to its publication. I will present the most important suggestions below.

[Figure]

Further, the paper needs a thorough checking by the authors as there are quite a few spelling mistakes and some not clearly written parts which are minor but still need correction or rewriting. I include more minor correction suggestions into the second part of the review.

Main points:

1. Add a paragraph in introduction giving some key references regarding the joined inversion methodologies.

2. Paragraph "3.1 Inversion" needs partial rewriting and clarification as it is not very informative. Both ERT and seismic data processing was made using same minimization scheme but the first paragraph of this section refers only to the geoelectrical parameters and ignores seismic. Further, the joined inversion approach is not clearly explained. It is understandable that part of joined inversion approach is published before but still the reader needs to get an idea of the procedure, esp. to know what approach and parameters were used in this work. Currently this occupies just 3 lines (p5,ln20-23).

3. The authors use the word "coverage" to describe the resolution of the inverted areas. Although I understand the purpose of using the word "coverage" I am not sure if this is the best term to describe the model resolution since I am not aware of the word "coverage" being used as a technical term (maybe is used in seismic imaging). Why not using the word resolution instead? If you decide to use the term "coverage" you need to make a clear definition of it prior to its use as it is different for the case of electrical images (sensitivity norm) and the seismic images (seismic ray density). No definition of it

4. In continuation to the above I consider that the use of the sensitivity matrix norm (not just a summation ! as mentioned in line 2 pg6 ) to get a metric of the resolution of the geoelectrical inversion cells is interesting however its direct use into the inverted geoelectrical results generates some problems:

- Shading generates complicated figures that non-expert readers are not easy to follow. In particular, strictly speaking, the color scale in ERT images is not fully correct as the presented rainbow color scale does not incorporate the overlaid alpha shading. For example, the reader cannot easily distinguish between highly shaded light green and yellow parameters so cannot really appreciate the actual resistivities behind the highly shaded (almost colorless) areas. In any case I would think that the inverted images somehow need to be presented as any resolution metric related image could be an additional one.

- As said the authors choose to use a binary (0,1) masking for the seismic data but a shaded one for the ERT images. Apparently they could have used the same approach for both: why not using seismic ray density, or seismic model resolution for alpha shading also to the seismic data? Or conversely why not using a 0,1 approach for the geoelectrical images: use a threshold sensitivity norm value below which all inverted resistivity values are blanked.

- A further consideration is that I have doubts if the sensitivity matrix norm is an indicative proxy for deciding the reliability of the inversion result of a particular parameter. Sensitivity norm values are also dependent on the size of the parameters. In this work, parameters have uneven sizes (coinciding with triangular elements) so small sized parameters may exhibit small sensitivity norm values also because of their size and thus they may be appearing as partly shaded. However, the geoelectrical images are the product of an inversion procedure which is being subjected also to structural regularization (smoothness). This regularization is an important factor of the inversion and effectively it operates in a way that increases the reliability of smaller parameters. This fact is not taken into account into the alpha shading which is solely based on the parameter sensitivity matrix values. In this context, I would think that the diagonal of the model resolution matrix would be a more reliable indicator for accessing the reliability of the inversion as it also takes into account the inversion procedure.

Overall, I agree with your idea to include a metric for evaluating the inversion results but

this need to be done in a way that is more reliable and easier for potential readership (i.e. engineers) to follow. I have mentioned some possible suggestions of how this could be done, and I am sure that the authors can come with even better ways to address this issue in the revised manuscript.

5. Joined inversion Results: the geoelectrical joined inversion result are at parts difficult to justify. The joined inversion result clearly constrains the thickness of the sediments and also helps identify a fracture zone. The geoelectrical image however also depicts very high resistivity values with high "coverage" at the bottom parameters of the image esp. at the left bottom part of the image. This is very different compared to the independent inversion results. More importantly why now the bottom left part of the inverted space (e.g. see parameter at x=100m, z=-125) is more reliable? How this is justified given that the seismic image information is inexistent for this region and generally this part of the model is well established that it has an inherently low resolution in geoelectrical surveys? I believe that the high 'coverage' of this part of the model is due to the size of the parameters in combination of their extremely high resistivity values which however are not reliable. I believe that the points made in previous remark hold also for this case so again I feel that it strengthens that argument that authors need to reconsider the way they present the results.

Minor Comments

Page1-ln13-15: too much detail and new terms in abstract, please revise.

Page1-ln15: replace "separated" with individual

Abstract: consider writing a more concise abstract

Page1-ln17: replace "unusually" with "very"

Page1-ln18: replace "The results" with "The joined inversion results"

Page1-ln18: replace "imaging some geologic" with "the imaging of important geologic"

Page1-ln18: add in "been not detected" "been either not"

Page2-ln5: maybe replace "dry" with "unsaturated"

Page2-ln10: "extreme points" not clear what you mean.

Page2-ln11-12: how is this statement on the finding fits in this part of the paper. Seems out of context

Page2-ln33: "field surveys" you mean "geophysical surveys"

Pag3-ln1: "numerical background": what do you mean? Is it geophysical propery background?

Page3-ln4: replace "disposal" with "disposal site" or "disposal facility" (same in ln11)

Page3-ln24: replace "analysis" with "analyses"

Page4-ln14: how you decide "wrong GNNS position" add some more info about electrode positioning.

Page4-ln31: replace "submarine" with "sea bottom"

Page6-ln2: simple summation of sensitivities will result in an artificially low sum as geoelectrical sensitivities can be also negative. Did you actually use the "norm"?

Page6-ln7: change "conductivity" to "resistivity" as given value is in Ohm-m.

Page6-ln8: replace "pure water is as well constant 1440m/2" with "saline water is also constant 1400 m/s"

Page6-ln16: consider rephrasing "We follow a strict 2D scheme" You mean: "In this work data were processed using a 2D inversion scheme" ?

Page6-ln26: replace "anomalies" with "anomalous bodies" or "perturbing bodies"

Page6-ln28-29: replace "incorporated by a rectangular cube" with "simulated by a rectangular prism" and "water cube" with "water modelling body"

Page7-ln2-3: replace "nose" with "noise". Not clear how you added noise. Is it both a noise on apparent resistivity and on the measured voltages? Please explain and justify.

Page8-ln11: explain somehow what you mean with "compensation artefacts" for non-inversion expert readers.

Page8-ln20: "alpha shading": not sure that all readers are familiar with this term. Please explain.

Page8-ln24: replace "resistive" with "resistivity"

Fig 1: Add an Arrow in survey line. Replace "scheduled" with "measured" in caption

Fig 3: in Fig is "brown" but in caption "red"

Fig 4: Axes impossible to read in figure due to the dark colors. Also please add labels of exact material properties directly on the prisms as no exact color scale is provided.

Fig 5: Delete "pure" in caption

---

## Author Comment (AC1) · 2 Mar 2017

We like to thank all the reviewers and the editor for their constructive critics that will help to improve the manuscript significantly. In most cases we have additions to the text, one case (going over several points of reviewer 2) we are going to add another figure about the use of different measures (coverage and resolution radius) for appraising model uncertainty. This will, along with the other changes, help to be a guideline to use geophysics in similar settings and thus be of wide interest for the non-geophysical readership of the journal.

In the following we have answered all points by (a) a direct response and (b) by the intended changes to the manuscript at the given position related to the original

manuscript.

Reviewer 1

1) Unfortunately, it is currently not clear, what practitioners and scientists can learn from their survey. The authors present data that was acquired under what they describe as difficult conditions, but it is unclear, what this really meant for the survey and results and what others should learn from it. Should seismics and ERT be standard methods to be used for site investigation under these circumstances or not?

a) As "difficult conditions" we can mention for resistivity the extreme variation in electrode coupling, the conductive water, electrical noise from the nearby nuclear power plant and possibly the 3D-effects on the resistivity modelling. For the seismic the gas bearing sediments are a complicating factor. We recommend the combination of resistivity/refraction methods as a standard tool for site investigations under these geologic conditions. It is clear that the methods complement each other well, and we have references from land. They also complement the traditional drilling well.

b) (P02L32) The main objective was the localisation and characterisation of fracture zones under challenging conditions, which are the extreme variation in electrode coupling, possible 3D-effects on ERT data and high acoustic damping due to gas bearing sediments.

Difficult conditions regarding ERT are explained on P4L07- P4L12. Add: Large variations of the contact impedance can be handled by the used instrument. Nevertheless, this can be problematic for other instruments.

(P10L18) Therefore, a combination of geoelectric and seismic refraction should be used as a standard tool for site investigations under geologic conditions which are similar to those presented.
* * *
2) In the introduction, the authors claim that they are testing geophysical methods to

improve planning for infrastructure projects.

a) This must be a misunderstanding. It is claimed that geophysics is gaining more attention, when it comes to the improvement of infrastructure projects. This is meant as a motivation and not as an objective.

b) (P02L03) Geophysical methods for site investigations gained more attention lately in order to integrate high resolution point information from boreholes, respectively.
* * *
3) Also, only one of the four fracture zones that is geologically known was found using the survey and there is only speculation, why the others are not found. Also, it is unclear, which of the shear zones would be most critical for infrastructure projects.

a) Two of the four fracture zones are merging towards the profile line, thus only three fracture zones might be distinguished. It just can be a speculation, why no contrast in the physical parameter was seen. According to an internal SKB report all fracture zones are partly conductive but different in size. As all fracture zones are hydraulically conductive, they should have been detected if the model resolution would have been sufficient, but a complicating factor is the highly conductive sediments with most likely higher contrast in resistivity that may mask the fracture zones.

b) (P03L28) According to Wikbert et al. (1991) all fracture zones are at least partly water bearing. They also gave a judgment of the fracture zones according to Bäcklom et al. (1990). Based on that, the most critical fracture zones along the measured profile is NE-1, which is judged as "certain". EW-3 is also judged "certain", but hydraulically of minor importance. NE-3 and NE-4 are judged as "certain" as well. Both consist of several one to a few metres wide subzones, of which some are open fractures that are hydraulically highly conductive. In general, the fracture zones NE-3, NE-4 and EW-7 is judged to be "probable" in a hydraulic sense (Wikberg et al. 1991).

(P09L31) Only the NE-1 fracture zone could be identified by this survey, although the

fracture zones NE-3, NE-4 and EW-3 are as well partly water bearing according to Wikberg et al. (1991). As shown in Figure 1, NE-4 and EW-7 are close to each other at the profile line, which means that they can be most likely not separated in the inversion results. The low resistive sediments are a complicating factor with most likely higher contrast in resistivity that may mask the fracture zones by reducing the resolution such that it is not sufficient to resolve the fracture zones NE-3, NE-4 and EW-7.
* * *
4) However, the strengths and weaknesses of the different methods in conditions of crystalline rock in combination with high-conductivity water are not clearly discussed.

a) While conducting underwater ERT parts of the current system will flow through the water, which can be included in the inversion. Thus only the loss of investigation depth is a weakness of the ERT method under these conditions. Seismic is unaffected by water as the refracted wave travels in the medium with higher velocity.

b) No change.
* * *
5) It is also not clearly discussed why the current study is needed and what it should add, compared to the existing ones.

a) This is a representative case study for the combination of geoelectric and refraction seismic in typical Scandinavian geologic conditions at a coastal region. There are references for combining refraction/ERT on land, but we didn't find a case study that combines refraction/resistivity in a marine environment. This survey was also a test for a joint inversion of ERT and seismic data. It shows the improvement of the results compared to a single method approach for these conditions.

b) We will rephrase the given explanation and add this to the manuscript. (Point 3-5 is mainly an improvement of the introduction)
* * *
6) Generally, the introduction should summarize the current state of research and discuss gaps that are addressed in the manuscript. Here, it is not clear, which gaps are being addressed and how.

a) The gap that we want to address is the methodical approach: joint inversion will help to get more reliable results compared to single method approaches. That shall make interpretation easier or at least more unique.

b) This will be pointed out more clearly. (End of introduction)
* * *
7) Furthermore, the introduction very much targets Sweden, while the study should be of more general interest to be published in an international journal.

a) Correct, we only referred to Swedish studies.

b) We will add some references for engineering projects in other countries that dealt with crystalline rock as hard-rock can be found in many countries.
* * *
8) The site description is a very short description of all the work that has been performed at the Äspö HRL. Unfortunately, information that would be important for the current study is missing: Previous seismic surveys in the rock lab shown? Was no surface seismic data acquired in the region? If other surface seismic data J. S. Kim, Wooil M. Moon, Ganpat Lodha, Mulu Serzu, and Nash Soonawala (1994). "Imaging of reflection seismic energy for mapping shallow fracture zones in crystalline rocks." GEOPHYSICS, 59(5), 753-765. doi: 10.1190/1.1443633 Cosma, C., Olsson, O., Keskinen, J., Heikkinen, P., 2001. Seismic characterization of fracturing at the Äspö Hard Rock Laboratory, Sweden, from the kilometer scale to the meter scale. International Journal of Rock Mechanics and Mining Sciences, ApplicationÂËŸa of Geophysics to Rock Engineering 38, 859–865. doi:10.1016/S1365-1609(01)00051-X Which seismic velocities are found in the rock lab for intact rock and how does it

change in fracture zones? What is the electrical resistivity of intact rock and of fracture zones? How likely are sedimentary deposits in the region? Have other studies or evidence on land shown extensive sedimentary deposits?

a) Actually, peer-reviewed articles about Äspö being relevant for this study (type of parameters etc.) are rare. Relevant previous results can mostly be found in SKB reports.

b) (P08L10) ... of about 5600 m/s, which agrees with the velocity for intact crystalline rock at Äspö HRL given in Wikberg et al. (1991). Brodic et al. (2016) showed recently that the velocity decreases from >5000m/s for intact rock down to 4200 – 4700 m/s. Brodic B., Malehmir A. and Juhlin C. ; Fracture System Characterization Using Wavemode Conversions and Tunnel-surface Seismics, Ext. Abstr., , EAGE Near Surface Geophysics, 2016
* * *
9) For the measurement techniques, some problems are described, but not really discussed how to be solved. For example: contact resistances vary by a factor of more than 1000. What does this mean for the measurements?

a) Large differences in the contact impedance (quality of the electrode grounding) can lead to oversteering of the signal, but this is handled by the used instrument which has input channels with automatic gain that can handle the full transmitter output signal on each individual channel. Since the input channels are galvanically separated one channel can have high gain and the next channel low gain without any common mode error problems. Unknown sharp boundaries between high and low resistivites are difficult to fit/image because smooth constrains of the inversion. But especially the last point can be included in the inversion by allowing sharp contrasts at known boundaries.

b) Answer already given in point 1).
* * *
10) For the positions: electrode positions were apparently measured using differential GNNS. How did this work for the underwater survey? What was the water depth? Does an accurate bathymetry model exist? Was this bathymetry model used, or was the multi-beam echo sounder used? Was the multi-beam echo sounder used to measure 3D ocean bottom topography?

a) Positioning is a very important point. An MBES survey was performed in order to map the sea floor topography very detailed. The result is a very accurate DTM (digital terrain model) that has been used in the modelling. Positions of sensors (E/S) are coincident and measured with sufficient accuracy. Water depths are small (<10m) and due to this a GNSS position of each shot point (20m spacing) has been enough.

b) The bathymetry model used for getting the heights for sensor-positions, but not for modelling.
* * *
11) With a difference in resistivity of more than a factor of 10000 between the water and the rock, it seems very important to know the position of the interface accurately. Were electrode positions deleted because they were in the wrong place, or simply, because the GNNS positioning did not work accurately?

a) The exact positioning is very important, but this was not a problem because of the precise Bathymetry measurements. Thus no electrode positions were deleted due to poor positioning.

b) (P04L08) A model based on accurate bathymetry measurements was used to determine the heights of the sensor positions at the seabed.
* * *
12) For seismic refraction tomography: Crystalline rock is a perfect target for reflection seismic surveys, as shown in many studies. Why is only refraction seismics being used here? Low velocity zones – especially if they are just thin fracture zones – are very

difficult to image in refraction tomography from the surface. Reflection surveys in the area (see references above) have shown the great potential of seismic surveys under these circumstances. If the potential of geophysical methods should be addressed here, why not process the data for both reflections and refraction and compare the results?

a) The depth to bedrock is one of the two main targets. This can't be given by P-wave reflection at these small depths. According to Cosma et al. 2001 reflection seismic was rarely used for crystalline rock especially for shallow depths. The wavelengths are too long and in a high velocity environment we need some distance before we get a proper reflection. This is of course what we get from the refraction seismic. Weakness zones and fractures are the other main target. The Kim et al. (1994) paper presents material that shows that these can be mapped by reflection seismic. However, there is no information in the top ~50m of the bedrock from reflection seismic. Another arguments against reflection in our case: Our study aims at urban environments. Reflection lines need to be much longer and it is unlikely that we can count on getting space for this in most cases. If we could, then there would be a much higher cost for performing reflection seismic. It is not common to get that kind of budget in Swedish infrastructure projects.

b) No changes in the manuscript.
* * *
13) The section on inversion is somewhat confusing and not fully consistent: For example, Equations 1a, 1b and 2 do not agree. Using eq. 2, the weighting matrix is missing in 1b. Also, according to the text, the model vector contains logarithmic resistivities. What about the slowness or velocity? Concerning the use of different norms: Most inversion schemes (and I think the one of Günther 2006b is not different) use a reweighting scheme to implement a L1 norm.

a) Equation 2 is an alternated version of $\Phi_{m}$ from equation (1a) and (1b) and

thus just extended with the weighting matrix $W_{c}$. That is the key point of this joint inversion. But this is a little bit inconsistent.

b) We will rewrite the paragraph and the second equation to make this clear. It will be mentioned explicitly that the logarithms of the slowness is used. We will rewrite the equations and remove the L1 norm in the text as it is confusing. We will also include some more more information about the joint inversion.
* * *
14) The discussion of the weighting matrix and the introduction of joint inversion on P5L15-22 is confusing and not well motivated. Why should a joint inversion be attempted in the first place? What could be the advantages here? What are the difficulties with single-method inversions that could be overcome by joint inversion?

a) This comment is related to point 5). This survey was designed in order to perform a joint inversion which should be tested. It was also mentioned that model ambiguities can lead to misinterpretations. These can be reduced by a joint inversion, because more data and more information (model of another method) will influence the quality of the inversion result. The presented results clearly support this statement, which is also discussed. (See also answer to 5)

b) Although it is already written in the manuscript, we will put more emphasize on this in the introduction. One sentence in section 3.1 → depth of penetration.
* * *
15) The reference of Günther et al. 2010 is rather difficult to access. Why not give the actual equation here?

a) Mathematics and physics of a joint inversion is not the scope of this paper, that's why an extended abstract is referenced and a more demonstrative explanation for the joint inversion was written.

b) No changes in the manuscript as this would only be a repetition of the mentioned

paper.
* * *
16) Synthetic study: It is interesting, but not surprising to learn from the synthetic study that 3D effects under these circumstances can be severe. The authors conclusion is that "ERT data gathered at the Äspö test site are contaminated" and that cautions should be used for the interpretation. However, if the conclusion is that the data is contaminated, the conclusion should be that one needs to take the 3D effects into account in the 2D inversion or perform a 3D survey. Simply saying that one needs to be careful in the interpretation is something one could say even without the synthetic study. I suspect that not only the island created artefacts, but also bathymetry changes in the area of the profile effect the results.

a) This is a quantitative measure for the 3D effects here and how they perform with parameter contrasts that we have in Äspö. It makes it more clear how strong those effects really are. Especially when non-geophysicists will read this paper, it helps to understand the difficulty by visualizing it.

b) No changes on the manuscript.
* * *
17) A more interesting synthetic study would be to compare the current density in the rock and in the water. With a resistivity factor of more than 10000, I would suspect that only very little current flows through the rock, which directly limits the sensitivity and makes small uncertainties in the water geometry a major problem.

a) True, but ERT is also driven by driving away the current, that's why you can better see high resistivity anomalies. The positioning of the electrodes is getting more important with an increased resistivity contrast. But as we have excellent control about the sensor positions for the underwater part by bathymetry measurements, this is not assumed to be a problem

b) Currently, we perform a synthetic study, where a couple of sensor positions are moved by 30-40 cm in z-direction while using with a resistivity contrast of a factor of 1000. Thus the importance of the positioning in combination with large resistivity contrasts can be seen.
* * *
18) Would it not be possible to use a 3D forward model that incorporates the 3D bathymetry and still invert for a 2D model along the profile? For the water, one could simply use the known water resistivity and for the rock far away from the profile, one could use the known rock resistivity. Would it not be possible to invert for one single water resistivity value? Is the measured resistivity that at the ambient temperature, or normalized to 25 degrees?

a) Using the bathymetry model could be done, but it is time consuming. It is possible, by setting the water to a single region, which allows only one resistivity for the whole region. But the resistivity of the water was measured with a minimised Wenner measurement using the Terrameter LS. Thus, no extra temperature correction is needed, as no conductivity meter was used. As the true water resistivity is known, it doesn't make sense to invert for this.

b) No changes on the manuscript for that comment as a detailed bathymetry would confuse the readers and provide no general conclusions about the strength of the 3D effects.
* * *
19) It might be co-incidence and actually makes geological sense, but the thick sediment deposit lies at the position with the greatest water depth and thus could be influenced by the water depth.

a) The topography shows that this is indeed the deepest point (relative), but the absolute depth in comparison to the investigation depth is not so high. Especially in relation to profile length. Seismic also validates this by a low velocity zone. So we consider a

geometric error very unlikely.

b) It is not planned to make changes according to this comment.
* * *
20) Results: How do the values compare to those expected from measurements at the HRL? How do they compare to the previous surface seismic lines? What else is known about the fracture zones?

a) Correct. We have velocities for the bedrock from a report, which agree with our bedrock velocity. There is an extended abstract regarding seismic measurements from Uppsala University, which gives some velocity information as well. We didn't found any reference data for resistivity. For the fracture zones, no velocity or resistivity information was found. Only chemical analysis of the fracture fluid and the filling material could be found, but those are not relevant for surface ERT and seismic measurements.

b) This comment is strongly related to comment 8). With the mentioned report and extended abstract as references this point can be answered as well.
* * *
21) Here geophysical results are presented, but without incorporating ground truth from e.g. boreholes.

a) According to Wikberg et al. (1999) there should be a borehole penetrating fracture zone NE-1. Up to this point we couldn't find geophysical reference data from them, only geological reference data. The tunnel gives excellent ground truth for the positioning of the zones, but no information about ground truth on depth to bedrock was found.

b) No changes on the manuscript. (Some sentences about the type of reference data?)
* * *
22) Are there any other indications of this sediment deposit? Are any drill holes available in the area? From other studies in the area: Are similar deposits known to exist? Are they likely, given the geological history?

a) Especially this sedimentary deposit was not known before (written in the text) and no other surface measurements were done that cover a lake. That's why no other deposits are known. A discussion with researchers from the geologic department and a SKB report supports that this scenario is geologically reasonable.

b) Dahlin et al. (2014) and Dahlin et al. (2016) also found sediments on the seabed near Stockholm. We will add these references that shows and supports the general opinion that sedimentary deposits on the seabed are typical for this region. Dahlin 2014 reference... Dahlin 2016 reference...

---

## Author Comment (AC2) · 2 Mar 2017

We like to thank all the reviewers and the editor for their constructive critics that will help to improve the manuscript significantly. In most cases we have additions to the text, one case (going over several points of reviewer 2) we are going to add another figure about the use of different measures (coverage and resolution radius) for appraising model uncertainty. This will, along with the other changes, help to be a guideline to use geophysics in similar settings and thus be of wide interest for the non-geophysical readership of the journal.

In the following we have answered all points by (a) a direct response and (b) by the intended changes to the manuscript at the given position related to the original

manuscript.

Reviewer 2

1) Add a paragraph in introduction giving some key references regarding the joined inversion methodologies.

a) Information for other joint inversions are indeed missing.

b) Additional paragraph in the introduction will be written.
* * *
2) Paragraph "3.1 Inversion" needs partial rewriting and clarification as it is not very informative. Both ERT and seismic data processing was made using same minimization scheme but the first paragraph of this section refers only to the geoelectrical parameters and ignores seismic. Further, the joined inversion approach is not clearly explained. It is understandable that part of joined inversion approach is published before but still the reader needs to get an idea of the procedure, esp. to know what approach and parameters were used in this work. Currently this occupies just 3 lines (p5,ln20-23).

a) First part is correct, seismic was not explicitly mentioned in the inversion section, thank you. For the joint inversion explanation we thought it would be nice not to go to much into detail and explain this more figuratively.

b) We will rewrite explanations for the flow chart diagram in a more demonstrative way and put in the basic equations for seismic inversion.
* * *
3) The authors use the word "coverage" to describe the resolution of the inverted areas. Although I understand the purpose of using the word "coverage" I am not sure if this is the best term to describe the model resolution since I am not aware of the word "coverage" being used as a technical term (maybe is used in seismic imaging). Why not using the word resolution instead? If you decide to use the term "coverage"

you need to make a clear definition of it prior to its use as it is different for the case of electrical images (sensitivity norm) and the seismic images (seismic ray density). No definition of it.

a) The coverage is explained in the text and is not really a resolution in terms of data/model resolution as it is the sum of the sensitivities, although it is widely used as such. The seismic coverage is a standardized coverage, which is either 0 or 1, depending if the ray travels through a cell or not.

b) An explanation for seismic coverage is needed. In general we will add a subsection (or paragraph) concerning resolution and coverage with a new figure (see figure below for ERT) that shows resolution radii and the coverage for a final Äspö model. Here a real measure based on the calculated model resolution can be compared with the coverage and also justify that the easy-to-retrieve coverage can be used as a rough estimator for model reliability. According to this we modify the alpha shading using 3 values (1: reliable; 0.5: less reliable 0: not reliable). Alternatively, we define several lines with resolution radii that are plotted in the final result. We also generate such a plot with coverage and resolution radius for seismics. This will enhance the methodology impact of the paper.
* * *
4) In continuation to the above I consider that the use of the sensitivity matrix norm (not just a summation ! as mentioned in line 2 pg6 ) to get a metric of the resolution of the geoelectrical inversion cells is interesting however its direct use into the inverted geoelectrical results generates some problems:

4.1) Shading generates complicated figures that non-expert readers are not easy to follow. In particular, strictly speaking, the colour scale in ERT images is not fully correct as the presented rainbow colour scale does not incorporate the overlaid alpha shading. For example, the reader cannot easily distinguish between highly shaded light green and yellow parameters so cannot really appreciate the actual resistivities behind the

highly shaded (almost colourless) areas. In any case I would think that the inverted images somehow need to be presented as any resolution metric related image could be an additional one.

4.2) As said the authors choose to use a binary (0,1) masking for the seismic data but a shaded one for the ERT images. Apparently they could have used the same approach for both: why not using seismic ray density, or seismic model resolution for alpha shading also to the seismic data? Or conversely why not using a 0,1 approach for the geoelectrical images: use a threshold sensitivity norm value below which all inverted resistivity values are blanked.

a) Good point. A gradual alpha shading for seismic would give a wrong impression, as the seismic wave either travels through a cell or not. Thus, a cell is covered (1) or not (0).

b) We will change the ERT figures by using only three values for the alpha shading as described in answer 3b).

–––––––––––––––––––––––––––––––––––––––––––––––––––––––––––––––––––––––––––––––––

4.3) A further consideration is that I have doubts if the sensitivity matrix norm is an indicative proxy for deciding the reliability of the inversion result of a particular parameter. Sensitivity norm values are also dependent on the size of the parameters. In this work, parameters have uneven sizes (coinciding with triangular elements) so small sized parameters may exhibit small sensitivity norm values also because of their size and thus they may be appearing as partly shaded. However, the geoelectrical images are the product of an inversion procedure which is being subjected also to structural regularization (smoothness). This regularization is an important factor of the inversion and effectively it operates in a way that increases the reliability of smaller parameters. This fact is not taken into account into the alpha shading which is solely based on the parameter sensitivity matrix values. In this context, I would think that the diagonal of the model resolution matrix would be a more reliable indicator for accessing the

reliability of the inversion as it also takes into account the inversion procedure. Overall, I agree with your idea to include a metric for evaluating the inversion results but this need to be done in a way that is more reliable and easier for potential readership (i.e. engineers) to follow. I have mentioned some possible suggestions of how this could be done, and I am sure that the authors can come with even better ways to address this issue in the revised manuscript.

a) We think that here a normalization by the cell size is meant. Right now this is done. Also the resolution radii are naturally normalized by the cell size. See also previous points.

b) Additional explanation of normalization by the cell size will be included.
* * *
5) I find the forward modelling very interesting and important as it helps evaluating the data in a much better way. However forward modelling is restricted only to ERT data and is not extended to reproducing a seismic model (with a geometry identical to the one used in the field) which could then be jointly inverted with the synthetic ERT data. This joined inversion synthetic model could demonstrate the superiority of the approach (or maybe also some limitations) under the given field conditions and measurement geometry. Note that this suggested addition is not meant to serve as a general synthetic model study, but as a unique model tailored to the actual survey conditions which will really help to better evaluate the actual real data results.

a) Refraction data are not much effected by 3D effects. Due to the first arrival picking, only signals are taken into account that took shortest way or travelled in the fastest medium. The small island next to the profile consists of the same bedrock as directly in profile line. Assuming the same velocity, the first arrival is still from the signal traveling in profile line, because the way is shorter. The small bay (water body) north of the profile would be a low velocity anomaly, which can be ignored, because this will not give a refraction, as an increasing velocity is needed for that.

b) We will add a reference for this.

—————————————————————————————————————————————————————

6) Joined inversion Results: the geoelectrical joined inversion result are at parts difficult to justify. The joined inversion result clearly constrains the thickness of the sediments and also helps identify a fracture zone. The geoelectrical image however also depicts very high resistivity values with high "coverage" at the bottom parameters of the image esp. at the left bottom part of the image. This is very different compared to the independent inversion results. More importantly why now the bottom left part of the inverted space (e.g. see parameter at x=100m, z=-125) is more reliable? How this is justified given that the seismic image information is inexistent for this region and generally this part of the model is well established that it has an inherently low resolution in geoelectrical surveys? I believe that the high 'coverage' of this part of the model is due to the size of the parameters in combination of their extremely high resistivity values which however are not reliable. I believe that the points made in previous remark hold also for this case so again I feel that it strengthens that argument that authors need to reconsider the way they present the results.

a) The alpha shading is based on the sensitivity distribution of the final model. If the final resistivity distribution is different, the sensitivity based coverage will be as well. Sensitivity doesn't take the electrode contact into account. That means as a low re-sistive zone channels the current and the signals, a high resistive anomaly will do the opposite and leads to larger penetration depth. The seismic does not cover the on-shore part, but it delivers the bedrock interface towards the ERT result, which also affects the onshore part of the ERT result.

b) This comment is solved updating the figures to binary alpha-shading.

————————————————————

[Figure]

[Figure]

**Fig. 1.** For comment 3) - Calculated resolution radii distribution and coverage for the final ERT
model